# Value of Right and Left Ventricular T1 and T2 Blood Pool Mapping in Patients with Chronic Thromboembolic Hypertension before and after Balloon Pulmonary Angioplasty

**DOI:** 10.3390/jcm12062092

**Published:** 2023-03-07

**Authors:** Fritz C. Roller, Armin Schüßler, Nils Kremer, Sebastian Harth, Steffen D. Kriechbaum, Christoph B. Wiedenroth, Stefan Guth, Andreas Breithecker, Manuel Richter, Khodr Tello, Werner Seeger, Eckhard Mayer, Gabriele A. Krombach

**Affiliations:** 1Department of Diagnostic and Interventional Radiology, Justus-Liebig-University Giessen, Klinikstraße 33, 35392 Giessen, Germany; 2German Center for Lung Research (DZL), 35392 Giessen, Germany; 3Department of Internal Medicine, Universities of Giessen and Marburg Lung Center (UGMLC), Institute for Lung Health (ILH), Cardio-Pulmonary Institute (CPI), 35392 Giessen, Germany; 4Department of Cardiology, Kerckhoff Heart and Thorax Centre, 61231 Bad Nauheim, Germany; 5DZHK (German Centre for Cardiovascular Research), Partner Site Rhein-Main, 60323 Frankfurt am Main, Germany; 6Department of Thoracic Surgery, Kerckhoff Heart and Thorax Centre, 61231 Bad Nauheim, Germany; 7Department of Radiology, Kerckhoff Heart and Thorax Centre, 61231 Bad Nauheim, Germany

**Keywords:** CMR, CTEPH, parametric imaging, T1 mapping, T2 mapping

## Abstract

Background: Parametric imaging has taken a steep rise in recent years and non-cardiac applications are of increasing interest. Therefore, the aim of our study was to assess right (RV) and left ventricular (LV) blood pool T1 and T2 values in patients with chronic thromboembolic pulmonary hypertension (CTEPH) compared to control subjects and their correlation to pulmonary hemodynamic. Methods: 26 patients with CTEPH (mean age 64.8 years ± 12.8 SD; 15 female), who underwent CMR and right heart catheterization (RHC) before and 6-months after balloon pulmonary angioplasty (BPA), were retrospectively included. Ventricular blood pool values were measured, compared to control subjects (mean age 40.5 years ± 12.8 SD; 16 female) and correlated to invasive measures (CI, mPAP, PVR). Results: In both, control subjects and CTEPH patients, RVT1 and RVT2 were significantly reduced compared to LVT1 and LVT2. Compared to control subjects, RVT2 was significantly reduced in CTEPH patients (*p* = 0.0065) and increased significantly after BPA (*p* = 0.0048). Moreover, RVT2 was positively correlated with CI and negatively correlated with mPAP and PVR before (r = 0.5155, r = −0.2541, r = −0.4571) and after BPA (r = 0.4769, r = −0.2585, r = −0.4396). Conclusion: Ventricular blood pool T2 mapping might be novel non-invasive CMR imaging marker for assessment of disease severity, prognosis, follow-up and even therapy monitoring in PH.

## 1. Introduction

Due to the outstanding success of parametric imaging techniques in the diagnosis of various cardiac diseases, parametric imaging—native T1 and T2 mapping and also quantification of extracellular volume (ECV)—has already found its way into diagnostic recommendations [1]. This has meanwhile favored, that parametric imaging is subject of multiple studies in patients with different etiologies of pulmonary hypertension (PH). So far, pathological elevations of cardiac native T1 time and extracellular volume (ECV) compared to healthy volunteers, promising correlations to functional and hemodynamic parameters and influences on patient outcome and survival have already been demonstrated [2,3,4,5,6,7,8]. Even more, parametric imaging seems to be suitable to investigate therapy effects—for example, of balloon pulmonary angioplasty (BPA)—on cardiac function and pulmonary hemodynamic in chronic thromboembolic pulmonary hypertension (CTEPH) [9]. The ongoing research with all these previous efforts and published results make it clear that parametric imaging as a component of non-invasive cardiac imaging has still arrived beside volumetric measures in patients with PH.

In addition to purely cardiac fields of application, non-cardiac tissues are also increasingly being examined by parametric imaging. For example, a very recent study by Guo et al. investigated the value of hepatic T1 mapping in patients with PH and demonstrated, that hepatic T1 values were predictive for adverse cardiovascular events [10]. Furthermore, Tilman et al. investigated right ventricular and left ventricular T2 blood pool values in patients with left-to-right shunting and showed that the “right-to-left-ventricular blood pool T2 ratio—RVT2/LVT2” might be a novel imaging biomarker for detection of shunts without the need for additive phase-contrast acquisitions [11]. 

In particular, the approach to investigate native T1 and T2 blood pool values and ratios is completely new and seems very interesting, because previous studies have shown that blood T2 is sensitive to the level of blood oxygenation and quantitative T2 Mapping as a novel, non-invasive method enables estimation of blood oxygen saturation [12,13]. Since blood oxygen saturation is reduced in patients with PH, it would be interesting to find out to what extent parametric imaging is able to distinguish between PH patients and healthy subjects as well as in the context of therapies. Therefore, the aim of our study was to investigate left and right ventricular native T1 and T2 blood pool values and their ratios in healthy controls and in patients with CTEPH before and after BPA in correlation to hemodynamic parameters.

## 2. Materials and Methods

### 2.1. Patient Population

A total of 26 consecutive CTEPH patients (15 female; mean age 64.8 ± 12.8 years ± standard deviation (SD)) were included in this retrospective cohort study from February 2014 to September 2017. Table 1 presents the comorbidities of the CTEPH patients. Cardiac magnetic resonance (CMR) imaging was performed in all patients for clinical reasons as part of the routine examinations before and after BPA. The primary diagnosis of CTEPH was based on the results of the imaging and hemodynamic workup, which included ventilation-perfusion scintigraphy (V/Q-SPECT), computed tomography pulmonary angiography (CTPA) or dual-energy computed tomography (DECT), right-heart-catheterization (RHC) measurements and conventional pulmonary angiography.

To reduce contrast media and magnetic field risks for healthy volunteers, the first 26 patients (16 female; mean age of 40.5 ± 12.8 years SD), who had undergone cardiac MRI for exclusion of myocardial inflammation between October 2022 and December 2022, were selected as control subjects. The control subjects were only included in the control group if both cardiac MRI examination and patient follow-up were completely unremarkable (normal heart size and normal heart function, absence of wall motion abnormalities, no sign of valvular heart disease, no signs of myocardial or pericardial inflammation, absence of pulmonary edema or pleural effusions, normal diameters of pulmonary trunk and aorta). Exclusion criteria and contraindications for CMR examinations were: kidney failure, incompatible metal or cochlear implants, known gadolinium intolerance and claustrophobia.

### 2.2. CMR Imaging

Imaging was performed on a 1.5 Tesla MRI system (Avanto, Siemens Healthineers, Forchheim, Germany) using a six-element phased array cardiac coil. The standardized imaging protocol comprised localizers, CINE imaging/steady-state free precession sequences (SSFP) in long-axis (2-, 3- and 4-chamber view), and short-axis (SA) orientation, late gadolinium enhancement imaging (LGE), and parametric imaging with native T1 and T2 mapping. LGE imaging was performed 12 min after contrast media injection (Gd-BOPTA; Multihance, BRACCO Imaging, Milan, Italy; dose 0.15 mmol/kg)

T1 mapping was performed as an optimized modified Look–Locker inversion–recovery (MOLLI; “3-3-5”) sequence was used. After performing of in-line motion correction, three T1 maps were acquired (basal, mid-ventricular, and apical SA). Imaging parameters were as follows: slice thickness 8 mm; spatial resolution 2.2 × 1.8 × 8 mm; 6/8 partial Fourier acquisition; field of view (FOV) 240 × 340 mm^2^; matrix 192 × 124; flip angle 35°; TR 740 ms and TE 1.06 ms; TI 100 ms and TI increment 80 ms; trigger delay 300 ms; inversions 3; acquisition heartbeats 3,3,5—11 images were acquired during 17 heartbeats.

T2 mapping was performed as a standardized T2-prepared SSFP single-shot sequence, as previously described [14]. A nonrigid registration algorithm was used to correct for residual cardiac and respiratory motion. Imaging parameters were as follows: slice thickness: 6 mm; TR 3 × RR; echo spacing 2.5 ms; flip angle 70°; T2-preparation times (T2p): 0 ms, 24 ms, 55 ms; FOV: 244 × 300 mm^2^ to 325 × 400 mm^2^ depending on heart rate; matrix 104 × 160; bandwidth 947 Hz/pixel; acceleration factor 2. 

### 2.3. CMR Analysis

LV and RV native T1 and T2 blood pool values were measured in regions of interest (ROI) at the basal or midventricular SA section. All ROIs were drawn carefully within the LV and RV to avoid measuring of partial volume-averaging artefacts and registration errors with gradual T1 changes at myocardial borders, due to including of trabeculae and papillary muscles. The lower limit of the measured blood pool ROI areas was defined as 100 mm^2^ and the upper limit as 300 mm^2^ to guarantee comparable and valid measurements for all CTEPH patients and controls. 

The T1 and T2 measurements were performed by two board-certified and experienced radiologists (12 and 27 years of cardiovascular imaging experience, respectively), and all T1 and T2 maps were of diagnostic quality (no patient had to be excluded). Both investigators were blinded to the patient demographics. At baseline, the first investigator performed all native T1 and T2 blood pool measurements and repeated the measurements after 14 days to assess intraobserver variability. The second investigator performed all native T1 and T2 blood pool measurements to determine interobserver variabilities. Postprocessing was performed by using the cardiovascular imaging software version 42 (Circle Cardiovascular Imaging, Calgary, AB, Canada).

### 2.4. Right Heart Catheterization

Standardized RHC was performed in all patients via the right internal jugular vein (6F sheath and standard Swan-Ganz catheter) within the pre- and postprocedural (6 month after final BPA procedure) diagnostic workup. 

### 2.5. Statistical Analysis

The statistical analyses were obtained by using PRISM statistical software (Graphpad Software Version 9.5, San Diego, CA, USA). Patient characteristics were expressed by mean ± SD. All data were checked for normal distribution using the Shapiro–Wilk test. In cases of normally distributed data, Student’s *t*-test was used, and for not-normally distributed data, the Wilcoxon signed-rank test (non-parametric) was applied. The correlation strengths were tested using the Spearman correlation coefficient r and interpreted according to Hinkle et al. [15]. According to Hinkle r > 0.3 was considered as a low correlation, r > 0.5 as a moderate correlation, r > 0.7 as a strong correlation, and r > 0.9 as a very strong correlation. To assess intra- and interobserver agreement The intra-class concordance correlation coefficient (ICC) was used to assess intraobserver and interobserver agreement. An ICC > 0.8 was defined as an excellent agreement. All results were tested at a 5% significance level. An alpha error of being less as 0.05 was accepted as statistically significant.

## 3. Results

Table 2 presents the RV and LV T1 and T2 blood pool values for CTEPH patients and control subjects. RVT1 and RVT2 were significantly lower than LVT1 and LVT2 in both, CTEPH patients (*p* = 0.0063 and *p* < 0.0001) and control subjects (*p* = 0.0002 and *p* < 0.0001). Figure 1 shows representative T1 and T2 Maps in a patient with CTEPH and T1 and T2 Maps in a control subject. Compared to control subjects, RVT2 was significantly lower in CTEPH patients (*p* = 0.0065), whereas LVT2, RVT1 and LVT1 differed not significantly compared to control subjects. In contrast to RVT1/LVT1, RVT2/LVT2 differed also significantly between CTEPH patients and control subjects (*p* = 0.0006). Table 3 presents the RV and LV T1 and T2 blood pool values of the CTEPH patients compared to the control subjects. 

Table 4 presents the functional and hemodynamic response to BPA therapy and Table 5 presents RV and LV T1 and T2 blood pool values of CTEPH patients before and after BPA. BPA leaded to significant increases of RVT2 and RVT2/LVT2 (*p* = 0.0048 and *p* = 0.0036), whereas LVT2, RVT1, LVT1 and RVT1/LVT1 differed not significantly after BPA. Figure 2 shows RV and LV T2 values in a CTEPH patient before and after BPA. 

Interestingly, RVT2 showed a significant positive moderate correlation to CI before BPA (r = 0.5155, *p* = 0.007) and a significant positive weak correlation after BPA (r = 0.4769, *p* = 0.0138). Moreover, RVT2 also showed significant negative weak correlations to PVR before (r = −0.4571, *p* = 0.0189) and after BPA (r = −0.4396, *p* = 0.0246), whereas only non-significant weak correlations to mPAP before (r = −0.2541, *p* = 0.2014) and after (r = −0.2585, *p* = 0.2022) BPA were present. Table 6 presents the correlations of RVT2 and Table 7 RVT2/LVT2 ratio to CI, mPAP and PVR before and after BPA in CTEPH patients.

The intra- and inter-observer variabilities in CTEPH patients (intraobserver: T1LV r = 0.9503; T1RV r = 0.9879; T2LV r = 0.9118; T1RV 0.8907; interobserver: T1LV r = 0.9344; T1RV r = 0.9455; T2LV r = 0.9231; T1RV 0.9034; all *p* < 0.001) and control subjects (intraobserver: T1LV r = 0.9267; T1RV r = 0.9684; T2LV r = 0.9537; T1RV 0.9328; interobserver: T1LV r = 0.9165; T1RV r = 0.9453; T2LV r = 0.9233; T1RV 0.9277; all *p* < 0.001) were very low.

## 4. Discussion

Myocardial T1 and T2 times show excellent reproducibility and are reliably used in daily clinical routine in many cardiac diseases, for example, in myocardial inflammation [1], fibrosis [16,17,18], amyloidosis [19,20], iron overload [21] and Morbus Fabry [22]. In addition to the analysis of myocardial T1 and T2 times, T1 blood pool values before and after contrast material administration have also been well studied as in integral part of ECV quantifications [23]. In contrast, T2 blood pool values have not been studied so far and data is sparse. A first study on T2 blood pool values by Tilman et al. showed that LVT2 and RVT2 blood pool values and their ratio are suitable to diagnose left-to-right atrial shunting [11]. To the best of our knowledge, this is the first study in the field which investigated T1 and T2 blood pool values in patients with CTEPH compared to control subjects and in correlation to pulmonary hemodynamic.

The four most significant findings of our study are: (1)LV and RV native T1 and T2 blood pool values measured on basal SA sections are reproducibly in both control subjects and CTEPH patients.(2)RVT1 and RVT2 values are lower compared to LVT1 and LVT2 values in both control subjects and CTEPH patients.(3)Moreover, RVT2 blood pool values are significantly reduced compared to control subjects in CTEPH patients, whereas LVT2 blood pool values are similar.(4)Interestingly, RVT2 blood pool values show significant correlations to measures of pulmonary hemodynamic (mPAP and PVR) and correlates of oxygen saturation (CI) before and even after BPA.

To explain and understand these results, one needs to understand pathophysiological and technical MRI aspects. Relaxation times of blood are a function of hemoglobin quantity and oxygen saturation [24,25] and the magnetic susceptibility of hemoglobin depends on its oxygenation condition—oxyhemoglobin has a high signal on T2, while deoxyhemoglobin has a low signal. This means blood T2 times increases the higher the oxygen saturation and the intracellular hemoglobin content is, respectively; it could be shown that T2 alterations of blood oxygen saturation affects the blood T2 value [12]. 

In healthy people, the arterial oxygen saturation in the left heart is higher than the venous oxygen saturation in the right heart. This is concordant to our finding that LVT2 blood values are higher than RVT2 blood values in both control subjects and CTEPH patients. Compared to the control subjects, the RVT2 blood values are significantly decreased in CTEPH patients. This can be explained by the lower blood oxygen saturation and increased deoxyhemoglobin due to the increased PVR and right heart failure in patients with PH. Compared to that, the oxygen saturation in the left ventricle seems not affected as LVT2 blood values are similar between CTEPH patients and control subjects, again very well fitting to the pathophysiology of precapillary PH. Furthermore, RVT2 blood values were significantly positive correlated to CI and significantly negative correlated to PVR before and after BPA, whereas a weak not significant negative correlation to mPAP was present. These correlations impressively demonstrate that therapy-associated improvements in mPAP, PVR, and oxygen saturation are accompanied by concomitant changes in T2 values. At least, blood pool T1 mapping seem less affected by changes of oxygen saturation as they do not significantly differ between the CTEPH patients and the control subjects. This is in line with results presented by Lin et al., who showed that blood T1 times were independent of oxygen saturation [26]. The slight non-significant increases in blood pool T1 values in the CTEPH patients compared to the control subjects are most likely due to chance or combinations of influencing variables such as hematocrit and temperature, for example. Influencing variables and combinations of variables were investigated by Liu et al. in human neonatal blood at 3 Tesla [27].

The main limitations of this study are: First, it was a single-centre study with a relatively small number of included patients. Second, our study only included CTEPH patients and, therefore, only limited inferences on blood pool T2 values, oxygen saturation and pulmonary hemodynamic in other PH subgroups appear to be permissible. Third, postcontrast mapping, which might provide additional information, was not a subject of this study. Fourth, hemodynamic of the control subjects is not eligible as RHC is not obtained in healthy people. Fifth, hematocrit values, which also affect the measured native T1 and T2 blood pool times beside oxygenation, were not assessed.

## 5. Conclusions

Based on our results, we conclude that right ventricular blood pool T2 mapping might be a novel and additional non-invasive CMR imaging marker and increment for assessment of disease severity, prognosis, follow-up and even therapy monitoring in PH.

## Figures and Tables

**Figure 1 jcm-12-02092-f001:**
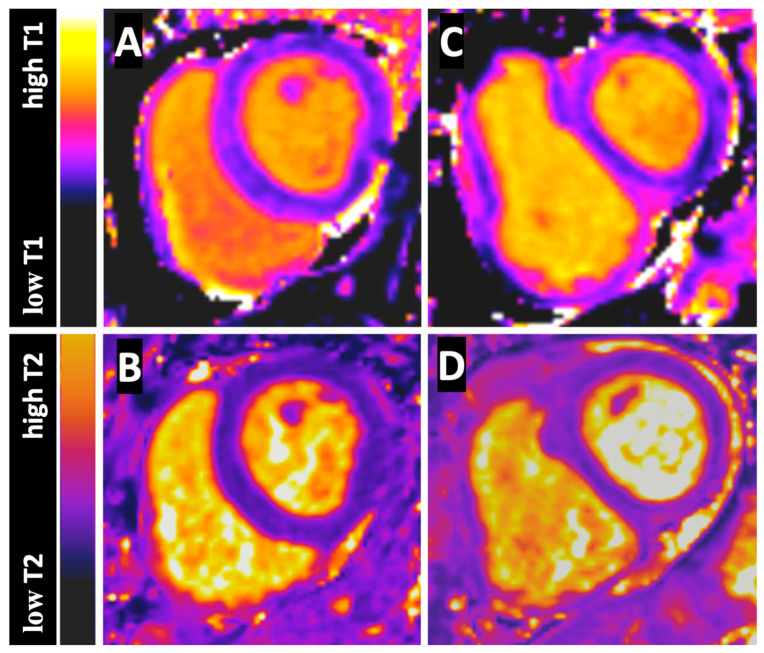
Figure shows the native T1 Map (**A**) and T2 Map (**B**) of a control subject and the native T1 Map (**C**) and T2 Map (**D**) of a CTEPH patient at basal short axis sections. The color-bar-scales on the left side presents the different T1 and T2 values from low to high.

**Figure 2 jcm-12-02092-f002:**
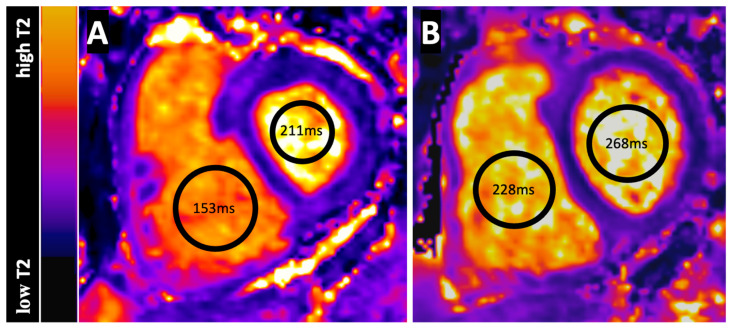
Figure shows the pre- (**A**) and postprocedural (**B**) T2 Map of a 53-year old male CTEPH patient at basal short axis section. T2 measurements were performed in regions-of-interest (black circles) for the left (LV) and right ventricular (RV) blood pool. The LVT2 and RVT2 values before BPA were 211 ms and 153 ms and after BPA 268 ms and 228 ms. Furthermore, a decreasing flattening of the interventricular septum is seen in the course of the BPA therapy. The color-bar-scale on the left side presents the different T2 values from low to high.

**Table 1 jcm-12-02092-t001:** Comorbidities of the CTEPH patients.

	*n* = 26
Age, years	64.8 ± 12.8
Female gender	15 (57.7)
Body mass index (BMI), kg/m²	24.7 ± 2.9
Diabetes mellitus	3 (11.5)
Arterial hypertension	16 (61.5)
Smoking	9 (34.6)
Coronary artery disease (CAD)	5 (19)
Atrial fibrillation	2 (7.7)
Chronic kidney failure	5 (19.2)
Glomerular filtration rate, mL/min	78 ± 20
Creatinine, µmol/L	0.95 ± 0.27
Cancer history	5 (19.2)
Chronic obstructive pulmonary disease (COPD)	1 (3.8)
History acute pulmonary embolism	22 (84.6)

Values are expressed as mean ± SD or as *n* (%).

**Table 2 jcm-12-02092-t002:** T1 and T2 values in CTEPH and control subjects.

	LV T1	RV T1	*p*	LV T2	RV T2	*p*
CTEPH	1493.1 ± 92.0	1460.2 ± 100.7	0.0063	237.4 ± 50.1	172.2 ± 35.8	<0.0001
Controls	1465.0 ± 64.5	1431.2 ± 71.7	0.0002	233.0 ± 38.4	199.8 ± 35.2	<0.0001

CTEPH—chronic thromboembolic pulmonary hypertension.

**Table 3 jcm-12-02092-t003:** CTEPH vs. control subjects.

	CTEPH	Controls	*p*
LV Native T1 time (ms)	1493.1 ± 92.0	1465.0 ± 64.5	0.2827
RV Native T1 time (ms)	1460.2 ± 100.7	1431.2 ± 71.7	0.3122
LV T2 time (ms)	237.4 ± 50.1	233.0 ± 38.4	0.8270
RV T2 time (ms)	172.2 ± 35.8	199.8 ± 35.2	0.0065
RVT1/LVT1	0.98 ± 0.03	0.98 ± 0.03	0.8809
RVT2/LVT2	0.72 ± 0.11	0.87 ± 0.13	0.0006

CTEPH—chronic thromboembolic pulmonary hypertension.

**Table 4 jcm-12-02092-t004:** Functional and hemodynamic response to BPA therapy.

	CTEPH Pre-BPA	CTEPH Post-BPA	*p*
RVEF	36.1 ± 11.6	50.0 ± 7.8	0.0001
mPAP	40.8 ± 7.3	31.3 ± 5.9	0.0001
PVR	532.2 ± 186.3	347.2 ± 104.2	0.0001
CI	2.6 ± 0.8	2.8 ± 0.7	0.08

RVEF—right ventricular ejection fraction, mPAP—mean pulmonary arterial pressure, PVR—pulmonary vascular resistance, CI—cardiac index.

**Table 5 jcm-12-02092-t005:** Pre- and post-procedural native T1 and T2 values and ratios in CTEPH.

	CTEPH Pre-BPA	CTEPH Post-BPA	*p*
LV Native T1 time (ms)	1493.1 ± 92.0	1518.3 ± 105.6	0.1482
RV Native T1 time (ms)	1460.2 ± 100.7	1480.4 ± 104.7	0.1375
LV T2 time (ms)	237.4 ± 50.1	253.4 ± 40.0	0.0678
RV T2 time (ms)	172.2 ± 35.8	196.7 ± 38.6	0.0048
RVT1/LVT1	0.98 ± 0.03	0.98 ± 0.03	0.6528
RVT2/LVT2	0.72 ± 0.11	0.78 ± 0.11	0.0036

CTEPH—chronic thromboembolic pulmonary hypertension.

**Table 6 jcm-12-02092-t006:** Pre- and post-procedural correlations of T2 values to hemodynamic parameters.

	R	CI 95%	*p* Value
RVT2 to CI			
Pre-BPA	0.5155	0.1483 to 0.7578	0.0070
Post-BPA	0.4769	0.09784 to 0.7351	0.0138
RVT2 to mPAP			
Pre-BPA	−0.2541	−0.5919 to 0.1596	0.2104
Post-BPA	−0.2585	−0.5950 to −0.1550	0.2022
RVT2 to PVR			
Pre-BPA	−0.4571	−0.7232 to −0.07270	0.0189
Post-BPA	−0.4396	−0.7126 to −0.05092	0.0246

CI—cardiac index, mPAP—mean pulmonary arterial pressure, PVR—pulmonary vascular resistance, BPA—balloon pulmonary angioplasty.

**Table 7 jcm-12-02092-t007:** Pre- and post-procedural correlations of RV/LVT2 ratio to hemodynamic parameters.

	R	CI 95%	*p* Value
T2 Ratio to CI			
Pre-BPA	0.06735	−0.3393 to 0.4528	0.7437
Post-BPA	0.07223	−0.7061 to −0.03788	0.7259
T2 Ratio to mPAP			
Pre-BPA	−0.1835	−0.5416 to 0.2309	0.3965
Post-BPA	−0.4290	−0.7061 to −0.03788	0.0288
T2 Ratio to PVR			
Pre-BPA	−0.2109	−0.5614 to 0.2037	0.3010
Post-BPA	−0.4326	−0.7084 to −0.04234	0.0273

CI—cardiac index, mPAP—mean pulmonary arterial pressure, PVR—pulmonary vascular resistance, BPA—balloon pulmonary angioplasty.

## Data Availability

The datasets used/or analyzed during the current study are available from the corresponding author on reasonable request.

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
