# Peer review of "Value of Right and Left Ventricular T1 and T2 Blood Pool Mapping in Patients with Chronic Thromboembolic Hypertension before and after Balloon Pulmonary Angioplasty"

_jcm, 2023, doi:10.3390/jcm12062092_

Round 1

Reviewer 1 Report

jcm-2231175

In this retrospective study, the authors studied T1 and T2 blood pool levels in patients with chronic thromboembolic pulmonary hypertension (CTEPH) compared to control subjects and in relation to pulmonary hemodynamics. This is a significant work that suggests for the first time that ventricular blood pool T2 mapping could be a unique non-invasive CMR imaging marker for assessing disease severity, prognosis, follow-up, and possibly therapy monitoring in PH. Nonetheless, from the standpoint of academic criticism, some technical issues must be resolved in order to improve the quality of this article, as listed below.

  1. This is a small sample size and a single-center study. The patient-to-patient variation is not well quantified, which jeopardizes the credibility of the data. Please add a table to show the demographic data for CTEPH patients and control involved in this study, especially listing the comorbidities of CTEPH patients which may affect the ventricular T1/T2 mapping results.

  1. Please remove the section below, it seems the authors keep the templet in this manuscript:

Line 279-280: “Supplementary Materials: The following supporting information can be downloaded at: 

www.mdpi.com/xxx/s1, Figure S1: title; Table S1: title; Video S1: title.”

  1. Please add in the protocol information, if not, revise the sentence accordingly. 

Line 289: “Justus-Liebig-University Giessen, Faculty of Medicine (protocol code XXX and date of approval).”

Author Response

Dear reviewer,

thank you very much for your positive feedback on our article and your suggestions for improvement. 
1. To address this point, we have included Table 1, which reflects the comorbidities of CTEPH patients.
2. The paragraph has been deleted - as suggested by you.
3. The protocol number of the ethics vote has been included, as you requested.
Thank you again for your quick review. We believe the changes further enhance our manuscript.

With kindest regards

Reviewer 2 Report

The manuscript entitled "Value of right and left ventricular T1 and T2 blood pool mapping in patients with chronic thromboembolic hypertension before and after balloon pulmonary angioplasty " aim to  to assess right (RV) and left ventricular (LV) blood pool T1 and T2 values in patients with chronic thromboembolic pulmonary hypertension (CTEPH) compared to control subjects and their correlation to pulmonary hemodynamic. It is an interesting topic and the manuscript was well written. However , We know T1 & T2 will be influenced not only by  oxygenation, but aloshematocrit. Could authors provide the data of hematocrit , or put into the limitation ?  

Author Response

Dear Reviewer,

thank you very much for your positive feedback on our article. 
Unfortunately, uniform current hematocrit values were not available for all patients. We have therefore included the issue in the imitations, as suggested by you.

Kindest regards

Reviewer 3 Report

Roller et al report novel marker in CMR for assessing CTEPH patients and show the value of ventricular blood pool T2 mapping as a marker of disease severity, as well as response after ballon angioplasty. 

This is a well-presented manuscript with conclusion drawn appropriately. Congratulate the authors on adding to literature on additional value of cMRI.  

One suggestion is that- while the authors conclude “ventricular blood pool T2 mapping might be 257 novel and additional non-invasive CMR imaging marker” however would urge the authors to be more specific. As in the study, LVt2 blood pool values were similar and only RVt2 blood pool values were significantly reduced in CTEPH patients. 

Author Response

Dear reviewer,

thank you very much for your positive feedback on our article and your suggestion for the summary. We have gladly taken up your point and included right ventricular in the summary.

With kindest regards